# A New Scheme for Ransomware Classification and Clustering Using Static Features

**Bahaa Yamany [1], Mahmoud Said Elsayed [2,\*], Anca D. Jurcut [2], Nashwa Abdelbaki [1] and Marianne A. Azer [3]**

1  School of Information Technology and Computer Science, Nile University, Cairo 12566, Egypt
2  School of Computer Science, University College Dublin, Belfield, Dublin D04 V1W8, Ireland
3  Computers and Systems Department, National Telecommunication Institute and School of Information Technology and Computer Science, Nile University, Cairo 12566, Egypt
\*  Correspondence: mahmoud.abdallah@ucdconnect.ie

**Abstract:** Ransomware is a strain of malware that disables access to the user's resources after infiltrating a victim's system. Ransomware is one of the most dangerous malware organizations face by blocking data access or publishing private data over the internet. The major challenge of any entity is how to decrypt the files encrypted by ransomware. Ransomware's binary analysis can provide a means to characterize the relationships between different features used by ransomware families to track the ransomware encryption mechanism routine. In this paper, we compare the different ransomware detection approaches and techniques. We investigate the criteria, parameters, and tools used in the ransomware detection ecosystem. We present the main recommendations and best practices for ransomware mitigation. In addition, we propose an efficient ransomware indexing system that provides search functionalities, similarity checking, sample classification, and clustering. The new system scheme mainly targets native ransomware binaries, and the indexing engine depends on hybrid data from the static analyzer system. Our scheme tracks and classifies ransomware based on static features to find the similarity between different ransomware samples. This is done by calculating the absolute Jaccard index. Results have shown that Import Address Table (IAT) feature can be used to classify different ransomware more accurately than the Strings feature.

**Keywords:** dynamic analysis; encryption; honeypot; Jaccard index; malware; machine learning; ransomware; similarity matrix; shared code analysis; static analysis

## 1. Introduction

Ransomware is the most trending malware. As per its dangerousness, it can be considered one of the significant internet industry threats [1]. Ransomware intends to obtain quick money from victims by encrypting systems and users' files until victims pay [2]. Ransomware works by encrypting data, files, and other system resources on the victim's computer and demanding a ransom in exchange for their release [3]. Ransomware is relatively simple to develop compared with other malware variants. At the same time, ransomware is challenging to deal with from a remediation aspect because once encrypted, the data causes significant losses for users and needs a significant amount of effort to undo the harm and return the system to its previous state [4]. The authors in [5] presented the lifecycle of ransomware. Figure 1 summarizes the different ransomware attack phases, from creation to extortion.

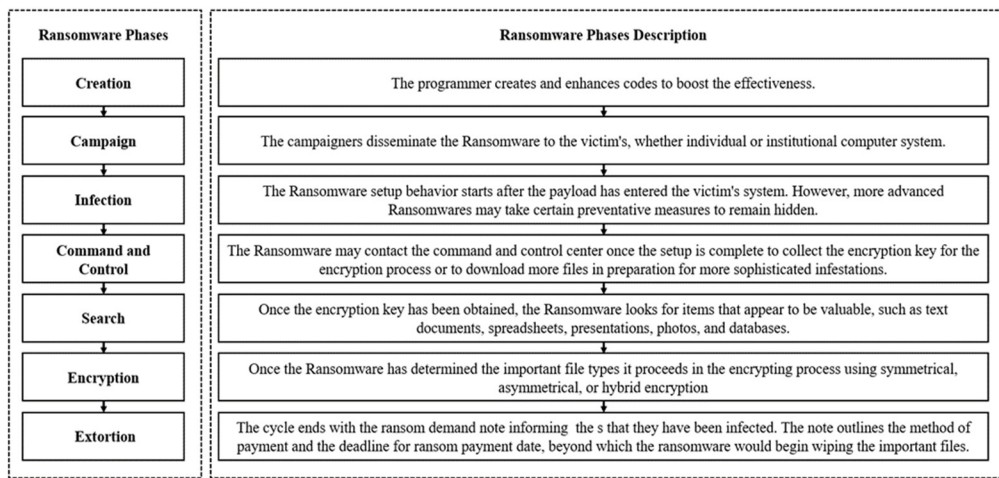

**Figure 1.** Ransomware lifecycle from creation to extortion.

*1.1. Malware via Ransomware*

Ransomware is a sub-category of malware designed to harm a computer or computer network. Ransomware can be defined as "an ever-evolving form of malware designed to encrypt files on a device, rendering any files and the systems that rely on them inoperable."

Malicious actors will then demand a ransom to decrypt the data. If the ransom is not paid, ransomware criminals frequently target and threaten to sell or disclose exfiltrated data or authentication information.

Ransomware authors are highly technically skilled people. Ransomware is still evading detection and infecting companies' networks across all industries. Defenders must continuously improve their dynamic detections.

Ransomware is spreading fast, which increases the number of security incidents on computers. Ransomware detection is not easy, but adding the propagation feature to the malware increases the dangerousness, as it is not centrally controlled. The ransomware industry has become more and more attractive from a business perspective, especially for governmental or political purposes. It offers many funds, encouraging ransomware authors to invent new evasion techniques against the ransomware detectors. Ransomware analysis is about understanding ransomware functions, determining their nature and purpose, host and network indicators, and persistence mechanisms. Ransomware analysis is the study or process of determining a malware sample's functionality, origin, and potential impact. Malware analysis can be divided into two main types:

i.     Static Analysis

Analysis of ransomware that is performed without the actual execution of the malicious code. Static analysis scales well and can provide better coverage of a ransomware binary code. However, static analysis can produce false execution behavior as code paths may not be reachable during actual execution [6].

ii.    Dynamic Analysis

Executing ransomware in an instrumented or monitored manner garners more factual information on behavior. Dynamic analysis can provide more accurate information on the actual execution behavior of a ransomware binary, and dynamic analysis can be computationally expensive [7].

The authors in [8] compared the static and dynamic ransomware analysis. Table 1 summarizes the comparison using the following parameters: Speed, safety, ability to analyze obfuscated and polymorphic hardware, level of false positives, and accuracy.

**Table 1.** Comparison between static and dynamic ransomware analysis.

| Parameters | Degree | Static Analysis | Dynamic Analysis |
|:---:|:---:|:---:|:---:|
| Speed | Low | | √ |
| | High | √ | |
| Safety | Low | | √ |
| | High | √ | |
| Obfuscated and polymorphic ransomware analysis | Not Able | √ | √ |
| | Able | | |
| False Positives Level | Low | √ | |
| | High | | √ |
| Accuracy | Low | | √ |
| | High | √ | |

### 1.2. Ransomware Tracking Approach

Analyzing a new ransomware sample can take effort and time to find the ransomware family that the sample belongs to. This could be achieved by running it through a multi-engine antivirus scanner such as VirusTotal. However, depending on this approach sometimes becomes difficult because VirusTotal detection naming may also have generic naming, as malware authors could check the VirusTotal database. Hence, they become aware that their ransomware is detectable, so they can change the ransomware's code or encryption function, which we try to track [9]. The second approach is to run the ransomware sample in CuckooBox or another malware sandbox to acquire a limited report on the callback servers and the behavior of the malware sample. This takes processing time, which is not practical in large malware datasets because it will take too much time to analyze the submitted ransomware samples dynamically [10]. The third approach we followed in our paper is ransomware shared code analysis tracking or similarity check analysis. This method compares two ransomware samples by calculating the proportion of recompilation source code they have in common and calculating the absolute Jaccard Index of the ransomware samples. This is done to find similar samples to track their encryption function, which allows us to apply existing decryption tools on similar samples to a reference sample that already has a decryptor.

### 1.3. Paper Contributions

This paper surveys the different ransomware detection approaches and techniques proposed in the literature. It also analyzes the various criteria, parameters, and tools used in the ransomware detection ecosystem. It presents the different recommendations proposed in the literature for ransomware prevention. The paper proposes a flexible and automated approach to extract malware's static features performed in a virtualized environment. Similarities and differences between malware features are computed, which allows malware classification and the detection of the similarity between different samples' static features. This helps our proposed system track the ransomware samples based on their functionality. Moreover, it aids in identifying similar ransomware samples to our reference sample. The second main goal after finding the similarity is to apply the ransomware's decryptor across similar ransomware samples to identify the ransomware encryption weaknesses in different ransomware families. The reference ransomware sample is SALAM ransomware [11].

The contributions of this paper are as follows.

1. Compare different ransomware families' infection behavior and provide a timeline for the ransomware's history.
2. Survey and compare the different ransomware detection approaches and present in detail the ransomware detection ecosystem, which comprises the detection environment, data analysis, Machine Learning, outcomes, evaluation criteria, and detection tools.

3. Study ransomware encryption mechanisms and techniques and identify the encryption algorithm used by SALAM ransomware.
4. Suggest a reference file that could be inserted into the environment's machines to be used later as the most significant reference file in case of encryption. The most prominent reference file is the non-encrypted copy of the encrypted ransomware file.
5. Set up an automated technique to extract different samples' features (Strings and Import Address Table) to calculate the Jaccard Index and Similarity matrix.

The remainder of this paper is organized as follows. Section 2 is a background about different static features used for malware tracking. Section 3 presents the effort done in the literature for ransomware detection. Section 4 presents our system setup results and analysis. Finally, conclusions and future work are presented in Section 5.

## 2. Background

In this section, we define and present the features affecting ransomware tracking and introduce the different static features used for malware tracking. Section 2.1 introduces ransomware types and history. Section 2.2 Ransomware infection source routines. Section 2.3 discusses ransomware encryption mechanisms and compares different ransomware encryption techniques. Finally, Section 2.4 presents malware tracking using static features.

### 2.1. Ransomware Types and History

Ransomware has become ubiquitous. No matter how much we organize to rid the world of the ransomware plague, we find that ransomware is becoming more common, and threat actors are becoming more brazen. Companies are crumbling under the weight of these attacks.

Ransomware is the most dangerous type of malware. Ransomware can be classified into different types:

Crypto worm-based ransomware: This is legacy ransomware as it uses other malware capabilities to spread through the network, but most modern anti-malware block this type of ransomware [12].

Ransomware-as-a-Service (RaaS): The Idea of Ransomware as a Service (RaaS) is that cybercriminals can create a customized version of various types of ransomware for profit; 30% of the profits go to the service developer [13].

Automated Active Adversary targeted ransomware [14]: the most dangerous type of ransomware used in APT attacks. One example is Shamoon data wiper malware [15,16] which was used by the APT-33 group [17] to attack the Middle East and Europe, whether for commercial or military reasons.

Each ransomware type has key features, infection spreading techniques, such as Phishing emails, Stolen RDP credentials, and exploit kits, exploitation mechanisms used by the ransomware such as exploiting the SMB protocol, which the WannaCry ransomware used in 2017, and finally, the ransomware families for each type, Table 2 summarizes the comparison between different ransomware malware behavior types.

### 2.2. Ransomware Infection Source Routine

There are several sources of ransomware infection. The percentage of different ransomware infection vectors was mentioned in [18]. There are an automated ransomware infection routines, such as random phishing emails and drive-by downloads. The new infection routine known as human-operated ransomware, which is the result of a targeted attack by cybercriminals who gain access to an organization's on-premises or cloud IT infrastructure, gain administrative access, and then use that access to push ransomware to critical data via domain controller group policy or network shared folders.

Human-operated ransomware attacks frequently include stealing credentials and then using those credentials to move laterally within an organization and get administrative access to more accounts. It is possible for fraudsters to exploit security configuration

gaps during maintenance windows. The end goal is to release a malware payload into the system.

**Table 2.** Comparison Between Ransomware Malware Behavior Types.

| Ransomware Type | Crypto Worm | Ransomware-as-a-Service (RaaS) | Automated Active Adversary |
|---|---|---|---|
| Key Features | Is a legacy ransomware. | Allows cybercriminals with limited technical expertise to launch ransomware attacks. The malware is made available to them, resulting in lower risk and more significant profit for the ransomware's developers. | Is an opportunistic ransomware attack. |
| Spreading techniques | Standalone ransomware that copies itself on other machines is used. | Dark web communities who buy the ransomware and use Phishing emails to infect the targets. | Ransomware is spread by attackers who use automated tools to search the internet for vulnerable IT systems. |
| Exploitation | Exploits the vulnerabilities within the operating system itself. | Social engineering techniques are used to deceive victims into opening emails or malicious attachments attached to emails or via exploit kits. | Targets are vulnerable to brute-force password-guessing attacks. They are a sought-after entry point. although victims may assume they are being targeted. |
| Detection | Monitoring the shared folders and limiting access. Security vendors use signature-based anti-malware engines to detect this type of ransomware. | Intrusion prevention and detection systems can prevent and monitor exploit attempts. Email security products can analyze the email attachment before the user receive it. | Anti-malware products can detect ransomware, but user awareness is key to eliminating this ransomware type. Virtual patching products can be used to minimize the ransomware actors' capabilities. |
| Ransomware Family | WannaCry, Ryuk | Satan, Cerber, Netwalker, MacRansom, SALAM, Conti | Egregor, Stop |

The sources of infection for most ransomware are illustrated in Figure 2 and can be summarized as follows in Table 3.

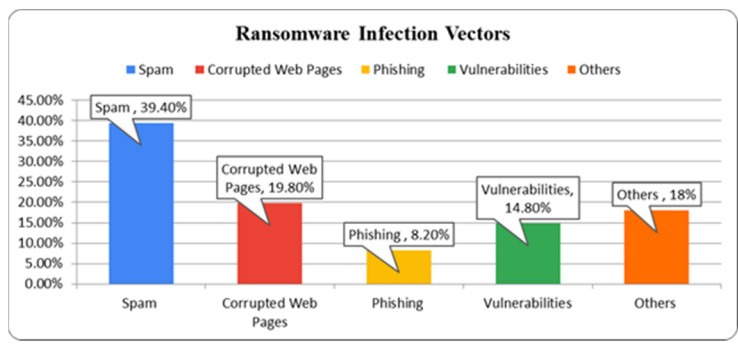

**Figure 2.** Ransomware infection vectors.

**Table 3.** Ransomware infection source types.

| Infection Source | Type |
|---|---|
| Random phishing emails | Automated legacy ransomware |
| Targeted phishing emails (Spear phishing) | human-operated ransomware |
| Leaked credentials | human-operated ransomware |
| RDP brute force | human-operated ransomware |
| APT attacks | human-operated ransomware |
| Vulnerable internet-facing systems | human-operated ransomware |
| Drive-by downloads | Automated legacy ransomware |
| Exploit kits | human-operated ransomware |

The authors in [19] summarized the ransomware evolution as shown in Figure 3 [19], while in [20], the authors discussed the history of ransomware in detail. In Table 4, we summarized the ransomware timeline with the date of the ransomware family and the damage caused by those ransomware families.

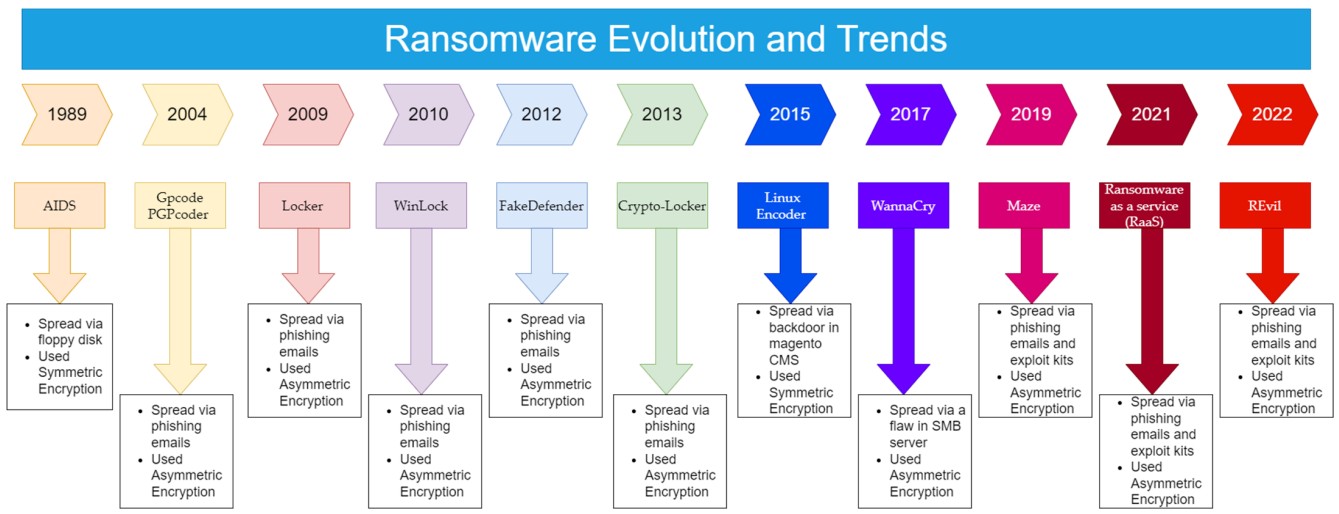

**Figure 3.** Ransomware evolution and trends timeline [19].

**Table 4.** Ransomware History Timeline.

| Date | Ransomware Family | Event Description | Damage |
|---|---|---|---|
| January 1989 | AIDS | AIDS Trojan emerges | Leakage of boot directories, loss of system files |
| December 2004 | Gpcode, PGPcoder emerges | The first quasi-modern crypto-ransomware called PGPcoder emerges | Loss of system and user files |
| November 2005 | Krotten emerges | The first known operating system locker called Krotten emerges | Data and money loss |
| March 2006 | Archiveus | The Cryzip crypto ransomware involving ZIP archives emerges | data and money loss |
| June 2006 | Cryzip | The first guides on how to create crypto ransomware appear in the underground | data and money loss |
| May 2009 | Krotten, WinLock | The Winlock ransomware for sale on underground forums for the first time | data and money loss |
| June 2009 | Locker ransomware | Sales of various locker ransomware surge on forums | data and money loss |
| July 2009 | Locker ransomware | Articles about developing locker ransomware appear in the underground | data and money loss |
| January 2010 | WinLock | The first locker ransomware affiliate programs emerge | Holding up the operating system, data and money loss |
| December 2010 | Crypto ransomware | Crypto ransomware returns with Encoder Builder, which is made freely available. | Holding up the operating system, data and money loss |
| July 2011 | Encoder | An improved version of Encoder is sold | Holding up the operating system, data and money loss |
| January 2012 | Reveton | Locker ransomware with the MBR overwrite functionality appears | Holding up the operating system, data and money loss |

**Table 4.** *Cont.*

| Date | Ransomware Family | Event Description | Damage |
|---|---|---|---|
| September 2012 | Encoder, Unlocker, Reveton, Citadel | A series of crypto-ransomware attacks start in Australia | Holding up the operating system, data and money loss |
| June 2013 | FakeDefender | The first crypto-ransomware affiliate program emerges | Holding up the operating system, data and money loss |
| September 2013 | FakeAV | The first CryptoLocker attacks and affiliate programs begin | Holding up the operating system, data and money loss |
| December 2013 | Crypto-Locker, PowerLocker | Known underground users report that the locker ransomware trade is dead and that the crypto-ransomware era has begun | Holding up the operating system, data and money loss |
| January 2014 | Simplocker, Locker, Koler, LockDriod, Cypto-Wall | Many new crypto-ransomware and affiliate programs emerge | Holding up the operating system, data and money loss |
| May 2015 | TorrentLocker, VaultCrypt, Tox ransomware, LowLevel04, Chimera, Linux. Encoder, CryptoWall | The publicly available crypto-ransomware called Tox emerges | Holding up the operating system, data and money loss |
| November 2015 | Ransomware as a service (RaaS) | The ransomware called Chimera is used to attack law firms only, and hackers threaten to publish stolen data. | Holding up the operating system, data and money loss |
| November 2015 | Linux Encoder emerges | The first Linux ransomware called Linux Encoder emerges | Holding up the operating system, data and money loss |
| December 2015 | Linux Encoder emerges | Many threads on underground forums are created in which threat actors discuss that only legal entities should be attacked | Holding up the operating system, data and money loss |
| February 2016 | Cerber | One of the most large-scale and notorious affiliate programs, Cerber ransomware, begin | Holding up the operating system, data and money loss |
| March 2016 | Crysis | The first macOS ransomware called KeRanger appears | Holding up the operating system, data and money loss |
| March 2016 | Locky | The notorious ransomware called Petya, with the MBR, overwrite functionality, emerges | Holding up the operating system, data and money loss |
| November 2016 | Cerber ransomware, KeRanger, Petya, Mischa, Satana, ZCryptor, CTB-Locker, Locky, TeslaCrypt | Some pieces of ransomware start using Telegram as a Command-and-Control (C&C) server | Holding up the operating system, data and money loss |
| May 2017 | WannaCry | Attacks use WannaCry ransomware, with an automatic spreading function as a worm | Holding up the operating system, data and money loss |
| June 2017 | Sopra, WannaCry, NotPetya | The ransomware NotPetya, which continued the WannaCry activity, emerges | Holding up the operating system, data and money loss |
| January 2018 | Zeus, GandCrab | The first modern ransomware affiliate program called GandCrab is born, and the targeting of legal entities begins | Holding up the operating system, deleting the backup data, data and money loss |
| March 2019 | GandCrab, REvil, Mephistophilu | The first RaaS called Snatch, which uses the double extortion technique, is released | Holding up the operating system, data and money loss |
| May 2019 | Maze | The ransomware called Maze is created | Holding up the operating system, deleting the Backup data, data and money loss |

**Table 4.** *Cont.*

| Date | Ransomware Family | Event Description | Damage |
|---|---|---|---|
| December 2019 | Ransomware Snatch, ChaCha/Maze, REvil, Babuk | The first DLS (Maze) appears | Holding up the operating system, deleting the Backup data, data and money loss |
| June 2020 | Ransomware as a service (RaaS), SALAM Ransowmare | The number of new RaaS affiliate programs surges | Holding up the operating system, deleting the Backup data, data and money loss |
| May 2021 | Ransomware as a service (RaaS) | "No more ransoms!": publishing RaaS on underground forums is banned | Holding up the operating system, deleting the Backup data, data and money loss |
| July 2021 | Ramp platform, Groove | Ramp a ransomware-related forum | Holding up the operating system, deleting the Backup data, data and money loss |
| January 2022 | REvil | REvil ransomware gang arrested in Russia | Holding up the operating system, deleting the backup data, data and money loss |

*2.3. Ransomware Encryption Mechanisms*

Some ransomware encrypt all discovered files and directories. Others, such as Cerber and Locky, look for and encrypt specific document files [21]. Others, such as Petya, simply encrypt the boot data and the file system tables. Some ransomware authors search for and encrypt cloud-based information, while others exclusively encrypt local files. It is impossible to predict which one will attack. Most ransomware encryption employs both public and symmetric key encryption [22]. Asymmetric key encryption is used to secure the symmetric keys that encrypt all files. The encryption process goes through the following steps:

1. The ransomware's author generates one or more asymmetric public/private key pairs and one or more symmetric keys. Each file or computer could have its symmetric key.
2. The symmetric keys are used to encrypt the data, and the plaintext data versions are permanently deleted once the encrypted version is completed. The encrypted data file copies are usually given a recognized file extension by most ransomware applications.
3. The plaintext version of the symmetric keys is erased after encryption with the asymmetric public key.
4. The ransomware's key storage server receives the asymmetric private key and waits for additional instructions.

Different forms of encryption are used by different ransomware authors, but they are all quite good and standard. The Maze ransomware group, for example, employs ChaCha20 or RSA with 2048-bit keys [23]. Many ransomware authors prefer to use the far more widely used AES encryption cipher, but ChaCha20 is much stronger and faster [24].

SALSA20 is an encryption algorithm used by the SALAM ransomware family to encrypt files [25]. We tracked the SALSA20 algorithm used by this family and found it vulnerable as it generates a random key to be used for encryption and uses it for all files on the infected machine. In each round of encryption, the key state is derived from the key and the previous key state. The derived key state is XORed with the current block of the file's content. The steps of changing the key state and XORing are repeated until all blocks of the file are encrypted; Obtaining the original key is not feasible as the key is 16 bytes and cannot be derived from the files or the remnants of the malware. The stream of the key state is obtained by comparing an encrypted file with its original unencrypted file from a backup. This yields the actual key stream, which was used to encrypt each file in the system. The key stream can be used to decrypt the files directly without requiring the encryption key. The key stream length determines how long the beginning of the file has been decrypted. The challenge would be to find for each machine an unencrypted file backed up in some place, and that file must be large enough to generate a large key stream

to decrypt other files in the system [26]. Table 5 summarizes the different ransomware encryption algorithms. The salsa20 block diagram is shown in Figure 4 [27].

**Table 5.** Ransomware Encryption Algorithms [28–30].

| Ransomware Family | Encryption Algorithm |
| --- | --- |
| SALAM | Salsa 20 |
| TeslaCrypt | AES-256 |
| Petya | Salsa 20 |
| Cerber | RC4 |
| Dharma | AES-256 |
| Shade | AES-256 |
| GrandCrab | RSA, AES |
| BadRabbit | Salsa 20 |
| Locky | RSA, AES |
| Darkside | Salsa 20, RSA |
| BlackMatter | Salsa 20 |
| Stop | AES-256 |
| Babuk | ChaCha |
| AgeLocker | ChaCha 20 |
| Conti | AES-256 |
| Maze | ChaCha 20, RSA |
| Ryuk | RSA, AES |
| Anatova | Salsa 20 |
| Snake | AES-256 |
| WestedLocker | AES-256 |
| GoldenEye | Salsa 20 |
| TorrentLocker | AES-CTR |
| Misha | Salsa 20 |

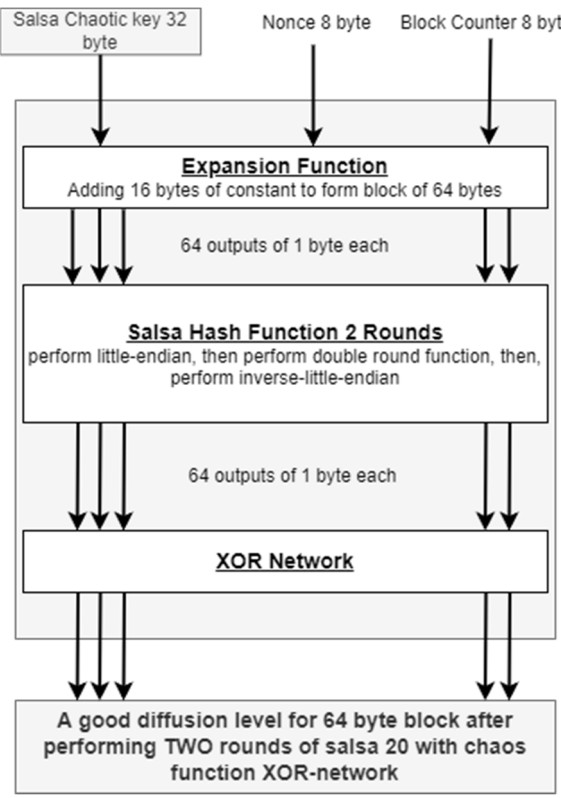

**Figure 4.** Salsa20 Block Diagram. Adapted with permission from Ref. [27].

### 2.4. Static Features Ransomware Tracking

Our proposed ransomware classification, clustering, and detection system in Section 4 aims to help ransomware analysts and reverse-engineers from redoing tedious tasks done before by other analysts and provide a joint collaborative analysis, estimating the amount of code shared by two malicious ransomware binaries before attackers assembled them.

There are a variety of approaches to this problem, but a consistent thread appears from the hundreds of computer science research papers that have been published on the subject to estimate the amount of shared code between ransomware binaries. We classify malware samples into different static features before comparing them; those static features could be strings, hashes, export, and import address tables [31]. Shared features between two malware samples are shown in Figure 5 [31].

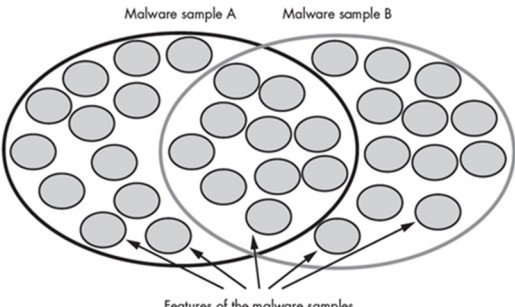

**Figure 5.** Shared features between two malware samples. Adapted with permission from Ref. [31].

An N-gram is a substring of a given text or speech string sample with a length of n. This string can include several types depending on the application. For example, it can include letters, words, phonetics, syllables, etc. N-grams are created by splitting a text string into substrings of fixed length. For example, world MALWARE 3-g will look like this "MAL," "ALW," "LWA," "WAR," "ARE." As a result of the string-based nature of analysis files, this technique has been widely adopted by security researchers to represent the features of ransomware [32,33]. We employ a similarity function with the following properties to determine the level of code commonality between two malware samples shown in Figure 6:

- It produces a normalized value that allows all similarity comparisons across malware samples to be compared on the same scale. The function should return a value ranging from 0 (no code sharing) to 1 (all code sharing) (samples share 100 percent of their code).
- The function should aid us in calculating accurate estimates of code sharing between two samples.
- Able to understand why the function does a good job modeling code similarities.

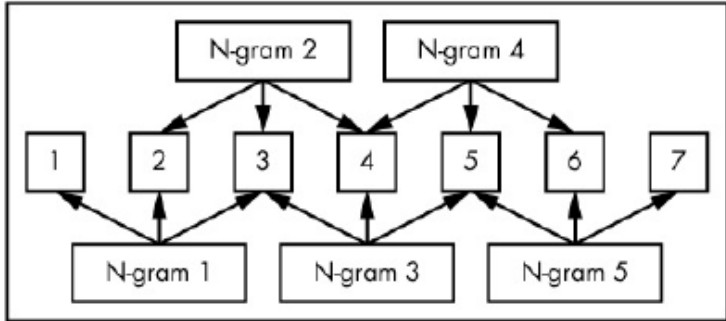

**Figure 6.** N-gram extracted from ransomware samples. Adapted with permission from Ref. [30].

The Jaccard index is a straightforward function with these characteristics. Even though alternative mathematical techniques to code similarity estimation (such as cosine distance,

L1 distance, Euclidean [L2] distance, and so on) have been tested in the security research community, the Jaccard index has emerged as the most generally adopted—and with good reason. It quantifies the degree of overlap between two sets of malware features simply and sensibly, providing us the percentage of unique features common to both sets normalized by the percentage of unique features in each group, JI = intersection length/union length, Jaccard Index explanation shown in Figure 7 [34].

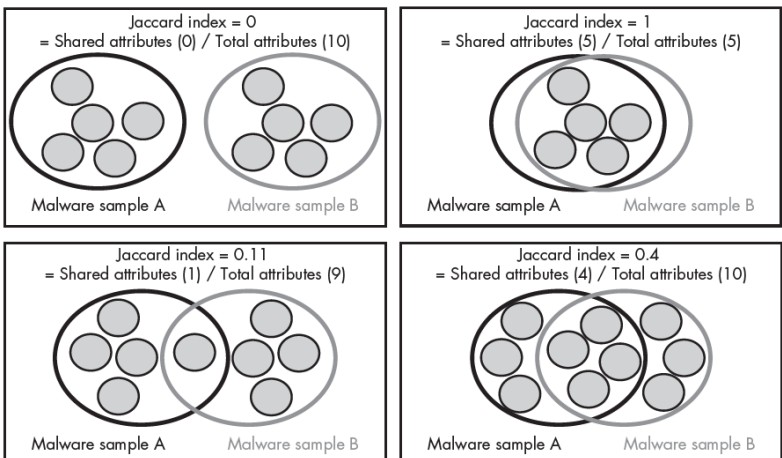

**Figure 7.** Jaccard Index between two malware samples. Adapted with permission from Ref. [30].

## 3. Related Work

Reverse engineering can be used for ransomware detection. Shared code analysis identifies samples that can be analyzed together (since they were generated from the same malware toolkit or are different versions of the same malware family), allowing us to determine whether a group of malware samples was created by the same developers. Given a new sample, a shared code estimate allows seeing which samples it is likely to share code with and what we know about those samples in seconds. Feature extraction, feature selection, classification/clustering, and decision are the four essential elements of a generic malware detection system. The raw data sample is fed into the feature extraction module, which extracts the most important qualities as a set of features. Following that, feature selection is used to alleviate the curse of dimensionality, minimize computational complexity, and improve system performance by quantifying feature correlations. A classifier/clustering scheme is given the generated feature vector. Finally, the decision module obtains a final binary decision: malware or benign [35]. This is only one technique of malware detection.

In this section, we present the effort made in the literature for ransomware detection and prevention. Ransomware detection approaches, ransomware detection ecosystem, and ransomware prevention recommendations are presented in Sections 3.1–3.3, respectively.

### 3.1. Ransomware Detection Approaches and Techniques

There are different classifications for ransomware detection approaches in the literature; one of them is categorizing the approaches to Machine Learning, Honeypot, and Statistical Analysis [5]. In the following, we present each approach, examples of the techniques that adopted the approach, as well as the advantages and disadvantages of each approach. It is worth mentioning that some ransomware detection techniques proposed in the literature use more than one approach.

#### 3.1.1. Machine Learning

Security vendors Kaspersky, Trend Micro, and FireEye use the machine learning approach to identify new malware samples. Machine learning algorithms frequently operate under the premise of constant data distribution or one that does not change over time by training the model to effectively reason any new sample in a test set once we

have a large enough training set. The model will continue to work as expected as time passes. After using machine learning to detect malware, we must acknowledge that our data distribution is not the same,

Thousands of software companies produce new types of benign executables that are significantly different from previously known types. Active adversaries (malware writers) are constantly working on avoiding detections and releasing new versions of malware files that differ significantly from those that have been seen during the training phase. Although there was no data on these types in the training set, the model must recognize them as benign. This leads to significant changes in data distribution and, in any machine learning application, the problem of detection rate degradation with time. This is a difficulty that cybersecurity firms that use machine learning in their anti-malware products must overcome. The design must be adaptable and allow for 'on-the-fly' model modifications between retraining sessions. Vendors must also have efficient systems for collecting and labeling new samples, regularly enriching training datasets, and regularly retraining models [36].

Machine learning algorithms can be classified as Bayesian, decision tree, dimension reduction, instance-based, clustering, deep learning, ensemble, neural network, regularization, rule system, and regression [5].

A technique to discriminate between ransomware and benign files, as well as malware, was proposed in [37]. The authors used Machine Learning methods to construct the detection model automatically, allowing new ransomware samples to be identified by developing the detection model. In [38], the authors examined significant research projects that used machine learning or deep learning to identify ransomware. An engine for ransomware detection using machine learning was proposed in [39]. It identifies and categorizes ransomware using a digital DNA sequencing engine and an AI machine learning network. It offers a technique of classification that distinguishes between different types of ransomware and groups them into well-known families based on their "digital genomes." With machine learning as the main focus, the authors in [40] analyzed and summarized the ransomware detection research status from different angles, such as sample acquisition, data preprocessing, feature selection, machine learning models, algorithms, and the evaluation of detection effectiveness. This classification methodology detected and categorized the detected ransomware into well-known families based on their "digital genome." The authors also evaluated the potential for future research into Android malware detection. Researchers in [40] used hybrid multi-level profiling to do a detailed forensic investigation of crypto-ransomware.

Furthermore, they used a unique behavioral chaining approach, as well as association rule mining and AI tools. Distinct behavioral chains were discovered during a hybrid multi-level inspection at the DLL, function call, and assembly levels, assisting in creating unique ransomware signatures and a specific dataset for the machine-learning model. Experiments have proven that the approach works with high accuracy and low false positives. With two class datasets, one of the machine learning algorithms had the best accuracy of 99.72 percent and the lowest false positive rate of 0.003. Experiments with multiclass malware families found a 94.6 percent accuracy rate, with an extremely low false-positive rate of 0.001. The chain ratio of ransomware behavioral profiling is a new concept.

### 3.1.2. Honeypots

Honeypots can identify the user with the number of edited files, which can inform actions. Honeypot principles revolve around gathering information about an attack and utilizing it for defense. User awareness training must be combined with email notifications sent to users, and the message may even ask them to remove their network cables. Therefore, using honeypots for ransomware detection is beneficial.

The authors in [41] used a combined approach; they benefited from machine learning in identifying malware by categorizing instances and used honeypots as a trap for packages that are suspected of containing malware. As classification methods, Decision

Tree and Support Vector Machine (SVM) were employed. In the study, the authors suggest architectural design as a malware detection method.

An Intrusion Detection Honeypot (IDH) was suggested in [42]. IDH was made up of three components: Honeyfolder, Audit Watch, and Complex Event Processing (CEP). It was designed to be attacked and serve as an early warning system to notify the user during unusual file operations. The authors in [43] proposed layers of deception mechanisms to identify any attempted hacking or ransomware using a deception strategy based on Honeyfiles and Honeytokens to obtain access to compromised private files. A honeypot-based strategy that uses machine learning methods to identify malware was proposed in [44]. A machine learning model was effectively and dynamically trained using data from an IoT honeypot as a dataset. Based on the client honeypot idea and active download interception, a study proposed in [45] a framework consisting of an IPS gateway, an analytic system, and a honeypot for identifying and detecting ransomware. Six modules made up the suggested structure. The IPS detects the download and routes it to the gateway. The static detector, dynamic detector, and honeynet evaluate the sample and establish its type and ransomware family. The notification module is in charge of informing and providing information to the user.

### 3.1.3. Statistics

Statistics may be used to analyze ransomware and better understand its key features. Statistical tests can detect unpredictability and may then be used to signal the existence of encryption is a common way of detecting ransomware [46]. Based on the frequency of opcodes in the portable executable file, the authors in [47] proposed an approach for detecting malware. The study used a machine learning system to detect false positives, false negatives, true positives, and true negatives in malware.

In [48], the authors created a similarity measurement algorithm-based malware detection technique. The proposed approach aimed to increase the speed and rate of malware detection. This method has several benefits, including a significantly higher speed because it uses opcodes directly and better detection results because it is unaffected by obfuscation and disassembly techniques.

The research in [49] offered a malware categorization method. This work suggests binary texture analysis over greyscale pictures generated straight from malware executables, which is motivated by the visual similarities among viruses from the same family. The method generates second-order statistical texture features over the visualized malware. This method is resistant to obfuscation techniques (e.g., packing, code relocation, and encryption). In [50], the authors evaluated five malware detection metrics without ground truth, a practical scenario that presents several technological difficulties. The ultimate objective was to create automated, principled techniques for assessing these indicators with the highest degree of precision. Concerning these five malware detection metrics, they provided statistical estimators. They employed fictitious data with established ground truth to verify these statistical estimators. Afterward, they used these estimators to measure five metrics using a sizable dataset obtained from VirusTotal and to quantify the five metrics. In [46], the authors examined the widespread usage of statistical methods currently used to identify ransomware, mainly focused on false positive rates. Their research's primary goal was to demonstrate how the present over-reliance on basic statistical tests in anti-ransomware programs may seriously compromise the accuracy and consistency of ransomware detection by often classifying samples incorrectly.

As mentioned earlier, this is only one way to categorize the different approaches used for ransomware detection. The techniques proposed in the literature can combine more than one approach. Table 6 summarizes the different ransomware detection approaches, their advantages, and their disadvantages.

**Table 6.** Comparison between ransomware detection approaches.

| Ransomware Detection Approach | Ref. | Description | Advantages | Disadvantages |
|---|---|---|---|---|
| Machine Learning | [34–40] | Machine learning (ML) uses data to identify patterns to build a model. The outcome may then be predicted using this model and fresh data [7]. However, the challenge with ML is finding the correct method to match the type of data and the desired result. | With sufficient training data, ML has the benefit of being able to anticipate the result with accuracy. Training data should be balanced in terms of the distribution of expected results. ML is less prone to obfuscation since it requires learning the pattern in the data. | A few rounds of trial and error may be necessary to find the best algorithm, which is frequently not simple. If sufficient caution is not used, biases and overfitting may also arise. |
| Honeypot | [42–45] | Setting up a honeypot allows the malware to target fake data. The ransomware can be found in these accessed files. | The honeypot files or traps may be set up, and they just need to wait for an assault. Consequently, the method does not need a lot of system upkeep or processing resources. | There is no assurance that a ransomware assault on the honeypot files will occur. Understanding the traits of the files the ransomware would target is crucial. The honeypot files or traps may be set up; after that, they must wait for an assault. Consequently, the method does not need a lot of system upkeep or processing resources. |
| Statistical | [46–50] | Statistics may be used to analyze ransomware and better understand its key features. | It is natural to investigate employing randomness tests to identify encryption (since the encryption process produces effective random data). The simplicity with which these randomization checks may be implemented could potentially be an advantage. | There may be major problems with the consistency and reliability of ransomware detection due to the over-reliance on simple statistical tests inside anti-ransomware solutions, which frequently result in incorrect classifications. Therefore, depending only on these basic statistical methods is insufficient to identify ransomware reliably. Instead, it is better to investigate higher-order statistics. |

*3.2. Ransomware Detection Ecosystem*

Recent ransomware research was presented in [5]. The authors surveyed the ransomware detection techniques proposed in the literature and included in their research the motivation methodology, results, limitations, and future directions of surveyed techniques. They also analyzed the ransomware detection techniques concerning many parameters such as the operating system for PCs and Mobiles, Cloud, Data Sources, different types of used machine learning algorithms, and outcome and evaluation criteria. Based on the studies in [5], we illustrated the ransomware detection ecosystem, including the different criteria and associated parameters in Figure 8. The number of literature occurrences of the ransomware detection parameters provided in [5] was used to create the comparative charts of the detection environment, data analysis, machine learning, outcomes, and evaluation criteria charts in Figures 9–13, respectively.

The authors in [51] categorized the ransomware detection techniques into Behavior-based, I/O Request packet monitoring, and Network traffic monitoring. They also provided a performance comparison of various ransomware detection techniques.

In [18], the authors surveyed and classified the ransomware detection tools and input information used by ransomware. Industry and academic researchers have proposed various methods for detecting ransomware, all of which involve gathering information from the malware in question before it is executed (or while it is running). This data is then utilized to determine if the piece of software in question is safe or harmful. The authors provide a categorization of the data and metrics gleaned from ransomware activity. They have something to do with the many forms of ransomware. In the first level of categorization, they grouped over 16 parameters related to prior actions into three groupings. They distinguished static or dynamic data gathered locally from the infected machine from data gathered through the network. In the parts that follow, they examine the following three groups: The information is locally static, meaning it is gathered prior to running

the malware program by extracting it. Second, data is gathered locally in real-time as the malware operates on the infected machine. Thirdly, information is gleaned through the malware's own network activity. In Figure 14, we summarize ransomware detection tools and input information used within the scope of [18].

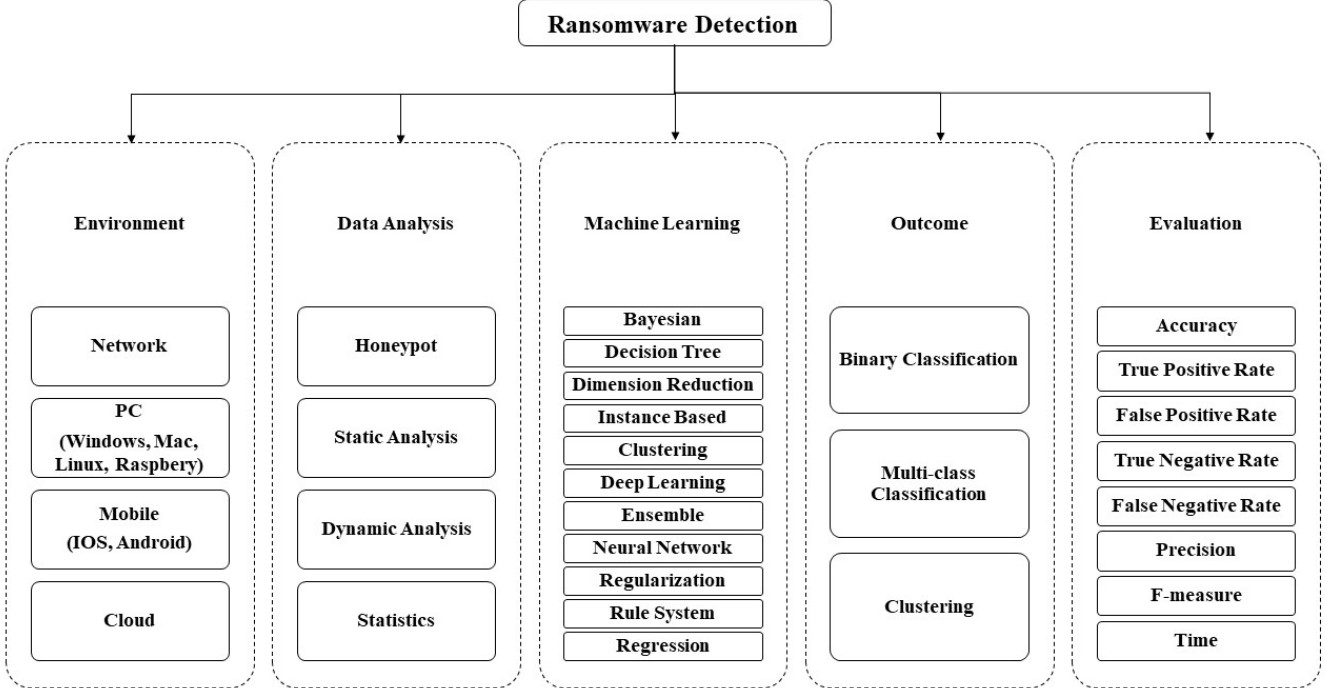

**Figure 8.** Ransomware detection ecosystem.

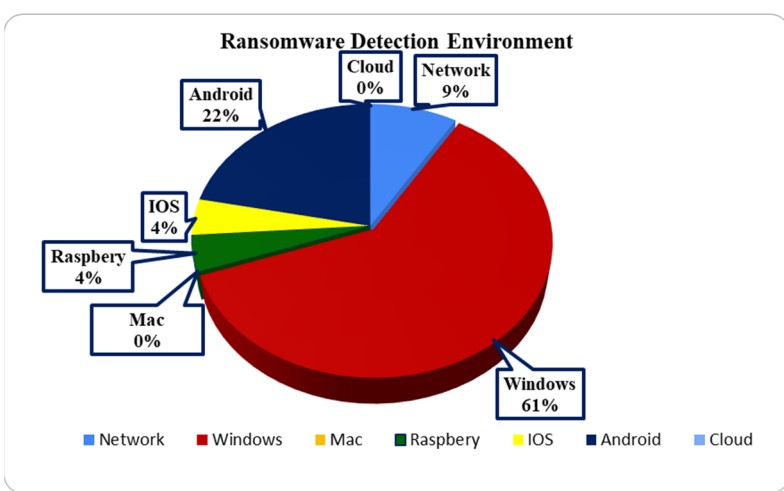

**Figure 9.** Different ransomware detection environments.

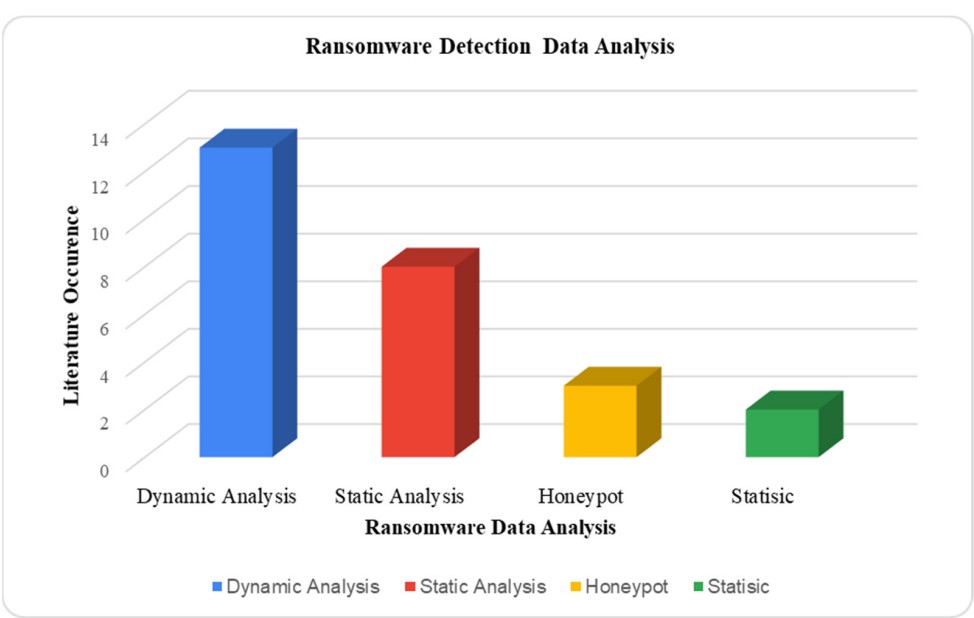

**Figure 10.** Data analysis approaches for ransomware detection.

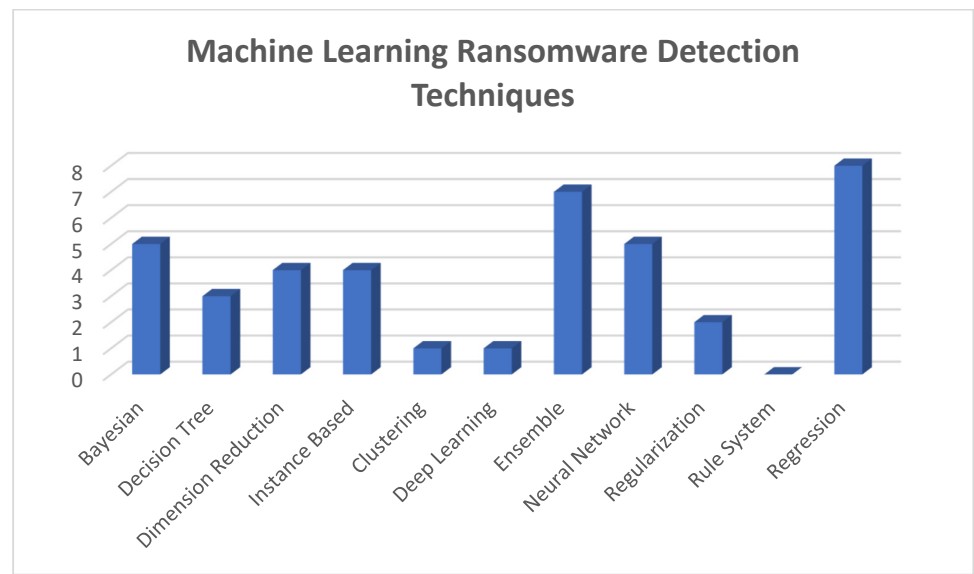

**Figure 11.** Machine Learning techniques used in ransomware detection.

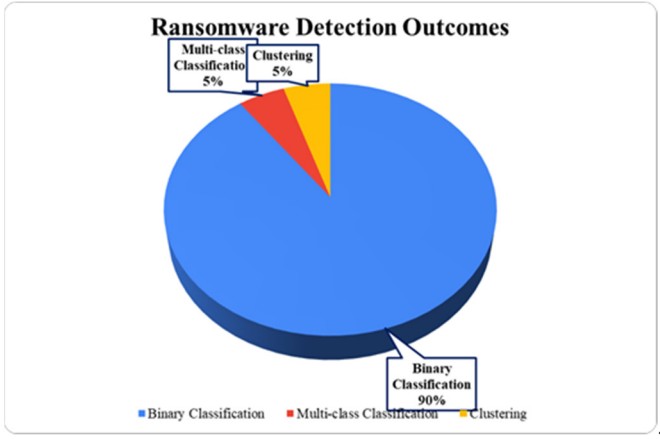

**Figure 12.** Ransomware detection outcomes.

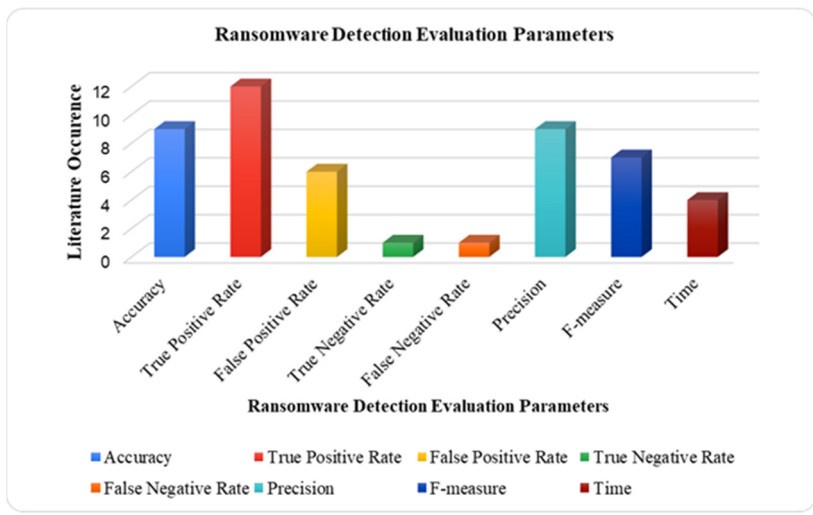

**Figure 13.** Ransomware Detection evaluation parameters.

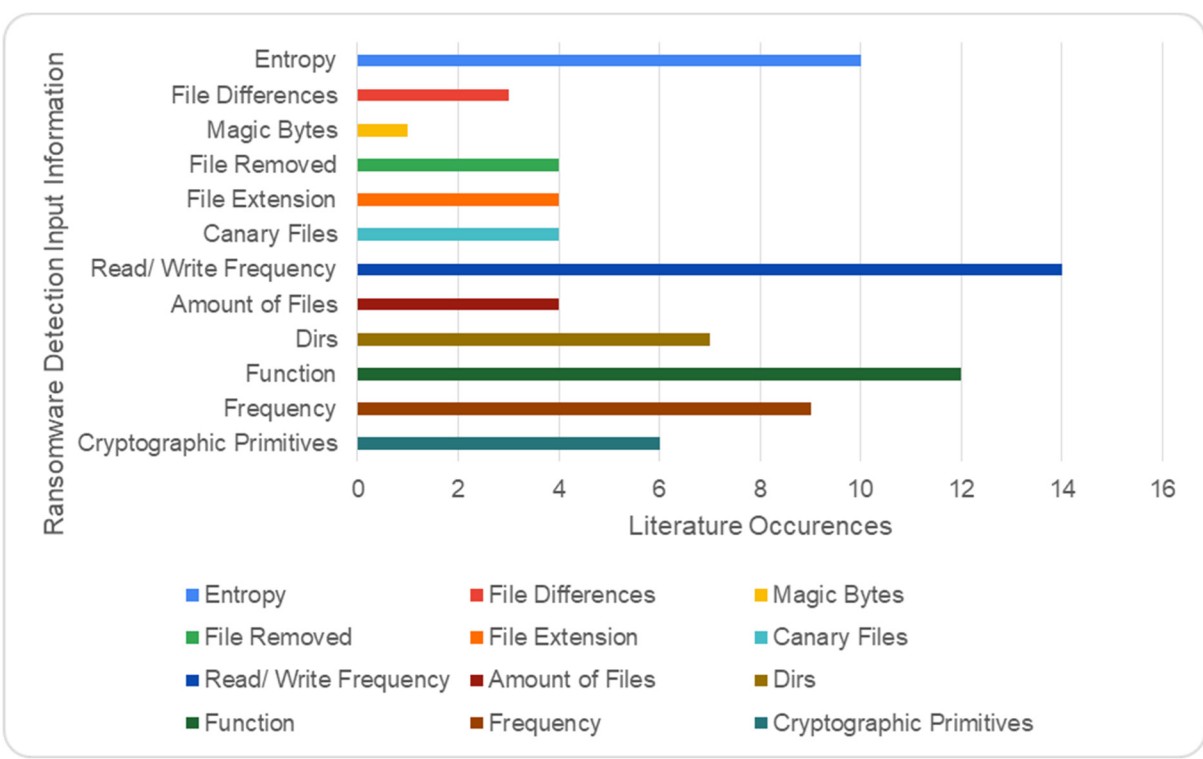

**Figure 14.** Ransomware detection tools and input information.

### 3.3. Ransomware Mitigation

Ransomware attack prevention is a challenging socio-technical issue [51]. An Approach to Preventing, Mitigating, and Recovering from Ransomware Attacks from a Socio-Technical Perspective related to the medical sector was proposed in [52]. This proposal is important nowadays, especially during the COVID-19 era. The approach consisted of four main pillars:

1.  First, health IT experts must effectively install and configure machines and the networks that connect them to guarantee adequate system protection.
2.  Next, healthcare companies must employ user-focused techniques, such as simulation and training on proper and thorough usage of computers and network applications, to enable more dependable system defense.

3. The firm also has to regularly monitor computer and program usage to spot suspicious activity, identify security issues, and fix them before they have a negative impact.
4. Lastly, enterprises must appropriately address ransomware attacks, recover promptly, and take steps to avoid them in the future.

The authors in [50] also proposed best practice solutions to prevent ransomware for both organizations and individuals. The following key recommendations were mentioned:

1. Backing up crucial data and making it easy to restore is one of the most effective lines of defense against ransomware attacks.
2. All software should be updated regularly.
3. Only a few people in the organization should have administrator accounts.
4. Backups should be verified and replicated offline.
5. Train employees.
6. Endpoint sandboxing, next-generation antivirus, and antivirus endpoint protection with updated signatures.
7. To prevent phishing attacks, use network sandboxing along with next-generation firewalls and email security.
8. Scale-out storage of the future, including capabilities for continuous data preservation and automatically collecting immutable snapshots.

In order to help individuals to prevent ransomware, the authors in [53] presented some suggestions. Figure 15 depicts their main suggestions.

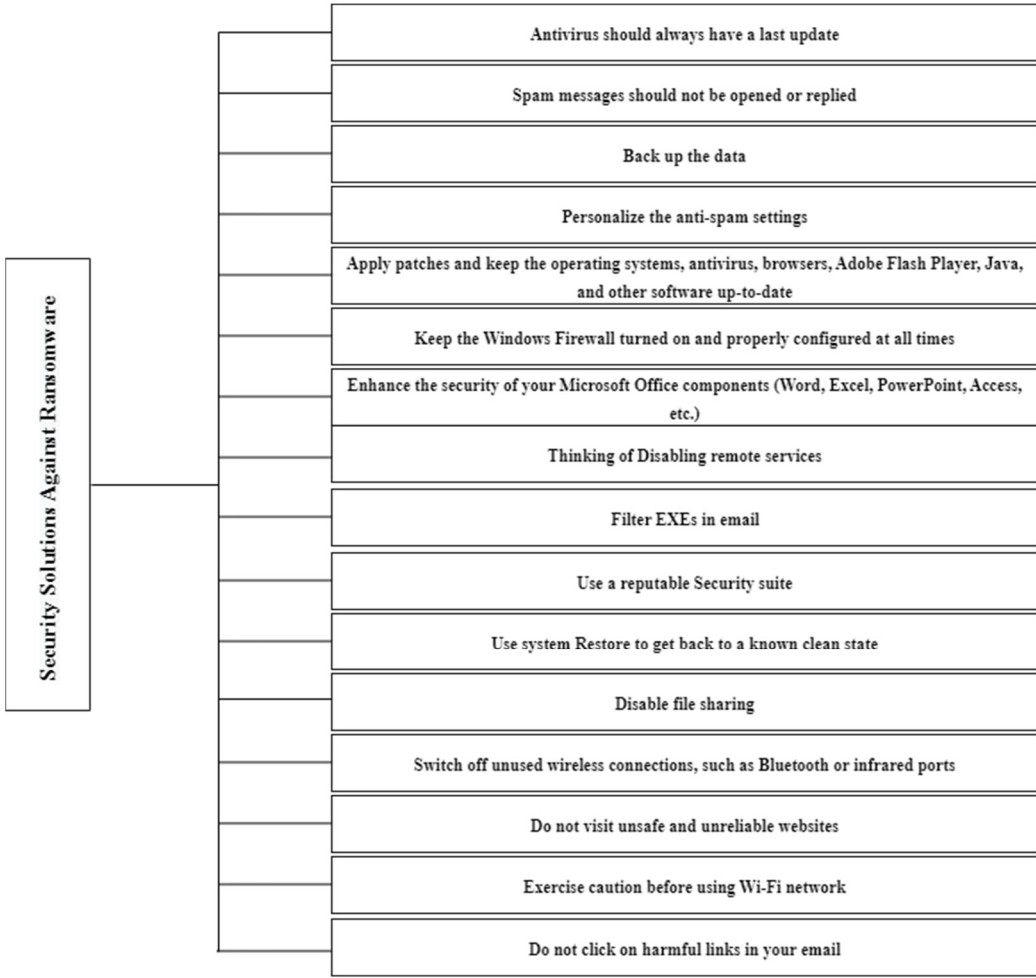

**Figure 15.** Security best practices to prevent ransomware. Adapted with permission from Ref. [54].

The authors in [55] provided suggestions for people and companies attempting to avoid a ransomware infestation. Figure 16 summarizes the different suggestions while

considering the extra miles that should be followed because ransomware developers try to bypass these precautions to fulfill their objectives.

| Mitigation Steps | Description | Ransomware's Tricks to Bypass Precautions | Extra Measures |
|---|---|---|---|
| **Back Up** | • If the data is backed up, there is no need to pay a ransom to get the data back. | • Some ransomware attempts to encrypt locally connected backup systems, | • It is also important to keep multiple backups |
| **Avoid Email Links and Attachments** | • Phishing attacks are the most common way to spread ransomware,<br>• Avoiding clicking on links or opening attachments in spam email will go a long way to avoiding ransomware. | • Criminals have also started using compromised advertising (malvertising) to spread ransomware. | • Turning off Java and JavaScript can also help.<br>• Business should train employees to avoid suspicious email.<br>• Corporate IT should consider standardizing ad blocking software. |
| **Patch and Block** | • The operating system, browsers, and security software should always be kept patched and up-to-date.<br>• Third-party plug-ins, like Java and Flash, need to be kept patched if they are allowed at all. | • Ransomware is constantly evolving to stay ahead of antivirus software | Software alone cannot be depended on to block an attack, other measures should be taken |
| **Drop-and-Roll** | • At the first sign of an infection, the infected machine should be immediately<br>• turned off (or unplugged) to minimize the damage to files.<br>• If it is connected to a network, administrators<br>• should immediately shut down the network to minimize the propagation of the ransomware | • Damage can aggravate due to human lack of awareness during infection | • Understand the Risks<br>• Develop Adequate Policies<br>• Institute Best Practices for Users. |

**Figure 16.** Ransomware mitigation for organizations and individuals.

## 4. Experimental Work

In this section, we present the experimental work done to study shared static features between different malware samples to check the similarity to a reference SALAM ransomware sample. The lab approach and tools used in our analysis are presented in Section 4.1, while the results are presented in Section 4.2.

### 4.1. Approach

Our approach is finding shared features between malware families to classify the ransomware samples first, then check the similarity between different samples by calculating N-gram and Jaccard Index and then applying the decryptor to similar samples. Jaccard Index has these properties: it is a simple function that does this. The Jaccard index has become the most popular way to figure out how similar two codes are, even through other mathematical methods. This is because it is the best way to figure out how similar two codes are. It shows how many malware features from two different sets overlap, giving us the percentage of unique features that are found in both sets normalized by the percentage of unique features that are found in each set to determine the degree to which the sample's bag of features resembles the bag of features from another sample. Similarity functions can help us determine how similar two malware samples are in terms of their source code

1. In this case, it gives a normalized value so that all comparisons between two malware samples can be put on the same scale. If the function does not share code, it should return a value between 0 and 1. (samples share 100 percent of their code).
2. The function should help us figure out how much code is shared between two samples (we can do this by doing experiments).
3. To understand why the function works so well, we should be able to quickly figure out why it works so well.

K means is a very popular algorithm for clustering objects. However, it requires the number of clusters k to be specified a priori. K means a prototype-based clustering, which means that a prototype represents each cluster; the prototype can be:

- Centroid: Average of similar points with continuous features.
- Medoid: The most representative or most frequently occurring point.

The choice of k is critical to the clustering performance.

N-grams: Contiguous sequence of n items from a given sample of text or speech (concept originating from the field of linguistics), "N-gram" is a small subset of some larger sequences of events with a predetermined length. Slide a window over the sequential data to extract this subsequence. This is done by going through a sequence of events and recording the subsequence from index i up to index i + N − 1.

We will use the following static features in our lab setup Strings and Import address table (IAT).

Our reference ransomware sample which we developed a decryptor before, its MD5 hash is "9c16b48fed1032c6bf6beb06e9d37fe2".

The ransomware classification and detection system submit samples through Python API and then applies a classification and clustering algorithm using disassembled binaries to generate mnemonic N-gram, calculating Jaccard similarity between different samples, and performing clustering on those samples to cluster them into classified clusters.

The proposed ransomware classification and detection system diagram is illustrated in Figure 17. It includes a System Controller with APIs for submitting ransomware samples to the static analysis server and querying the MongoDB NoSQL database for various properties. The Analyzer server on Windows retrieves the static characteristics and attributes from the given samples through the disassembler process.

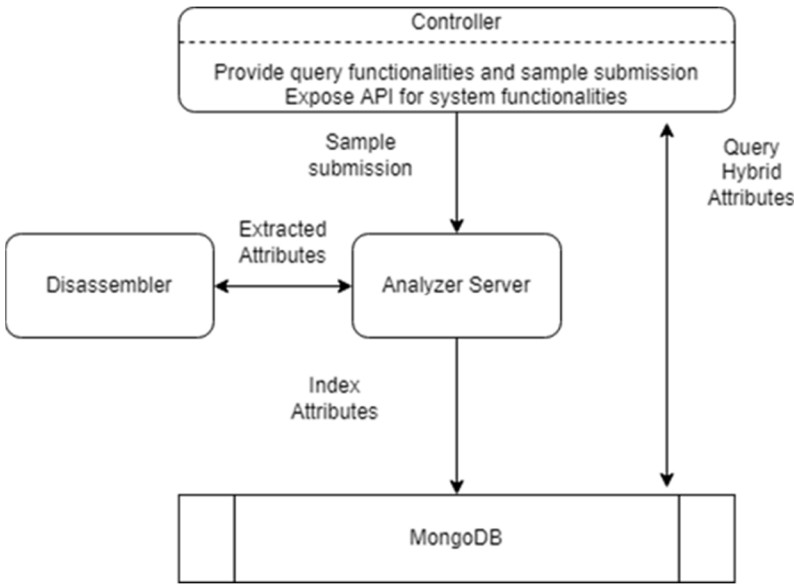

**Figure 17.** Static ransomware detection system block diagram.

The lab setup machines and tools used in our Lab are presented in Table 7. The controller is the key component that interacts with all the analysis engines and performs database queries to get relevant samples to the analyst's requests and submit samples to the analyzer server, which is responsible for disassembling executable binaries into a set of static features and extracting pertinent properties and features that the controller uses to categories the binaries. The disassembler is part of the analyzer server, which will disassemble the samples. MongoDB is used to index all extracted features to run the Jaccard Index familiarity on it. VirusTotal has its own clustering and similarity matching for each sample set. It includes a comprehensive graph view allowing users to rotate around malicious objects associated with the same campaign that helped us collect classified samples.

**Table 7.** Machines and Tools Used in Lab Setup.

| Machines/Tools | Description and Purpose | Techniques | Output |
|---|---|---|---|
| Controller (Windows 10) | The main component that interacts with all the analysis engines and performs database queries to retrieve relevant samples for analyst request. | Implement ML models | Hybrid attributes |
| Static Analyzer Engine (Windows 10) | Responsible for converting the executable binaries into a set of static features through disassembly to extract relevant static attributes used by the controller to classify the binaries. | Disassembler Decompiler | Control flow graphs Call graphs Functions Basic blocks Arguments count Cyclomatic complexity Instructions Strings Imports/Exports Segments |
| NoSQL MonogoDB database | Document-based store for all the extracted attributes by the analysis engines. | Receive controller queries for binaries hybrid attributes. | Hybrid attributes |
| Ghidra | Open-source disassembler used in samples disassembly and decomply in order to extract static features. | Decompiler Disassembler | Static features of the samples |
| Python PyCharm | Python IDE | Python | System scripts Database connections script Sample submissions script |
| VirusTotal | Open-source malware and URLs online scanning service. VT academic malware dataset ~10 GB of classified samples. | VirusTotal live hunt, Need the most recent ransomware samples | The used samples set collected over three months > 1000 ransomware samples spanning ~13 known classified families extracted |

*4.2. Results*

The below steps are used to find the similarity between the 16 samples of ransomware that used the Salsa 20 algorithm in their encryption. The first sample is the reference sample in which we developed a decryptor to break Salsa 20 algorithm encryption lifecycle.

Step 1—Store the samples.

Step 2—Index the samples.

Step 3—Search for samples.

Step 4—Visualize similarity.

We compared two static features to choose the best of them for our future work: the Strings and Import table.

4.2.1. Binary Strings Similarity

As shown in Figure 18, we found a similarity between our reference sample and one sample by 0.98749822, which is a good similarity score. The other samples are not similar to our reference sample. However, there are two matches in the families between the four samples into two families, as shown in Figure 19. As per the strings feature analysis, we have sixteen samples and thirteen different ransomware families, and one sample similar to the reference sample, samples with zero similarities Jaccard score are packed samples, and their strings are few compared with unpacked samples.

| MD5 Hash | Similarity Jaccard |
|---|---|
| ■ 9c16b48fed1032c6bf6beb06e9d37fe2 | 1 |
| ■ 41b0c21c7c32913214e67aeb0501bee5 | 0.001953125 |
| ■ 532abd14ecb22070d712db00cd3703c0 | 0.987498221 |
| ■ 153df2a5bb974c294ef5812280a297d0 | 0 |
| ■ 71b6a493388e7d0b40c83ce903bc6b04 | 0 |
| ■ 7e37ab34ecdcc3e77e24522ddfd4852d | 0 |
| ■ b06e2455a9c7c9485b85e9bdcceb8078 | 0.025390625 |
| ■ c830512579b0e08f40bc1791fc10c582 | 0 |
| ■ 596ebe227dcd03863e0a740b6c605924 | 0 |
| ■ 7b125a148ce0e0c126b95395dbf02b0e | 0 |
| ■ 0ed51a595631e9b4d60896ab5573332f | 0 |
| ■ fbbdc39af1139aebba4da004475e8839 | 0 |
| ■ e285b6ce047015943e685e6638bd837e | 0 |
| ■ 61139db0bbe4937cd1afc0b818049891 | 0 |
| ■ b14d8faf7f0cbcfad051cefe5f39645f | 0 |
| ■ 1a6d2bb4c752a458baf6fa2e56a62fcf | 0 |

**Figure 18.** Ransomware Binary Samples Strings Similarity Jaccard.

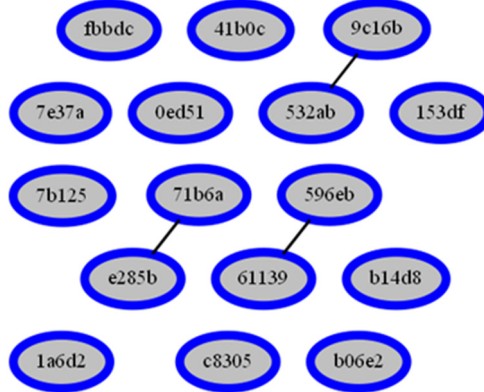

**Figure 19.** Ransomware binary samples Strings similarity graph.

4.2.2. Import Table

As shown in Figure 20, we found a similarity between our reference sample and one sample by 1, which is a full similarity score. The other samples are not similar to our reference sample. However, there are three matches in the families between six samples into three families, as shown in Figure 21. As per the strings feature analysis, we have

sixteen samples and twelve different ransomware families, and one sample similar to the reference sample.

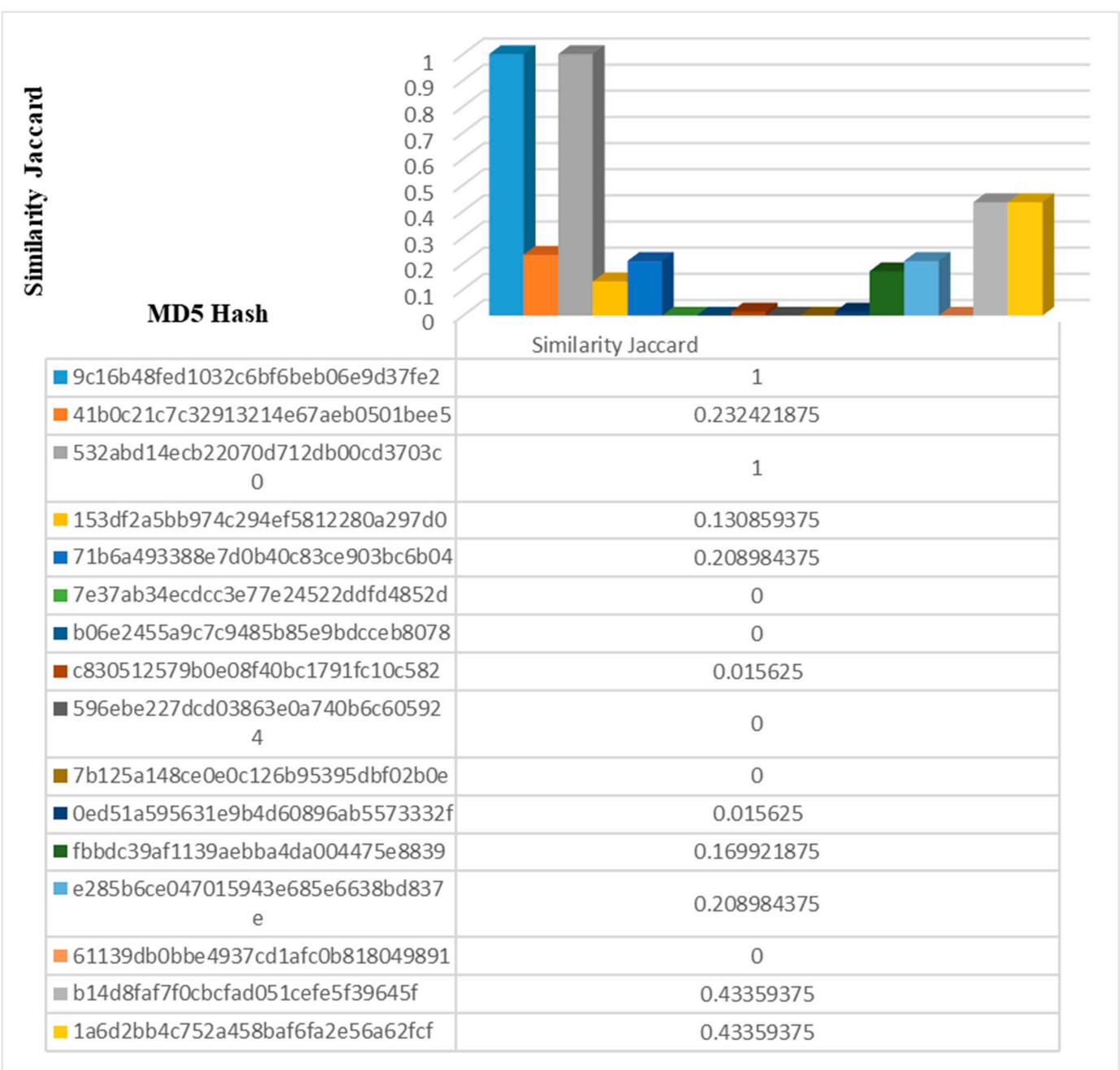

**Figure 20.** Import address similarity of ransomware samples.

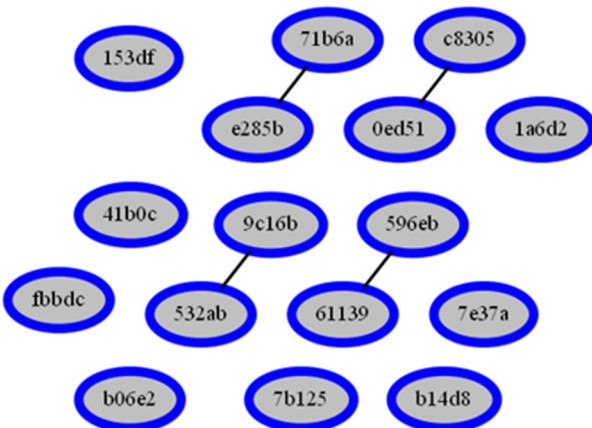

**Figure 21.** Import table similarity graph.

## 5. Conclusions and Future Work

The threat posed by ransomware to people's and companies' data files is increasing quickly. On an infected computer, it encrypts files and withholds the key to unlock them until the victim pays a ransom. Every year, hundreds of millions of dollars are lost due to this spyware. New versions constantly come out since so much money can be earned. In this paper, we surveyed the different ransomware detection approaches and techniques. We investigated the criteria, parameters, and techniques used in the ransomware detection ecosystem and the recommendations presented in the literature to mitigate and prevent ransomware attacks. In this paper, we provided an efficient malware indexing system that provides search functionalities, similarity checking, sample classification, and clustering. The system mainly targets native binaries, and the indexing engine depends on hybrid data from static analysis, comparing different ransomware families to find the similarity to a reference sample. We provided two solutions to previous research limitations to find the largest file in the infected machine. This file is compared with the encrypted sample to break the key stream to use it in decrypting other files from the same machine. We found that the import address table can be used as a static feature to classify different ransomware or malware families more accurately than Strings used as a static feature. The research limitation is the packed samples which the malware author obfuscates by hiding and packing the malicious code. Many strings exist in unpacked samples. However, obfuscated or packed samples contain few strings compared with unpacked samples. Therefore, the strings static feature similarity check is more accurate in identifying packed samples. In the future, we plan to develop a dynamic analyzer using sandboxing to analyze the packed samples and get more features from both static and dynamic analyzers to classify, index, and find similar samples.

**Author Contributions:** Conceptualization, B.Y., N.A. and M.A.A.; methodology, B.Y. and M.A.A.; software, B.Y.; validation, B.Y., N.A. and M.A.A.; formal analysis, B.Y., N.A. and M.A.A.; investigation, B.Y., M.S.E. and M.A.A.; resources, B.Y.; data curation, B.Y. and M.A.A.; writing—original draft preparation, B.Y., A.D.J., N.A. and M.A.A.; writing—review and editing, B.Y., M.S.E., A.D.J., N.A. and M.A.A.; visualization, B.Y., M.S.E., A.D.J., N.A. and M.A.A.; supervision, A.D.J., M.S.E. and M.A.A.; project administration, B.Y., M.S.E., A.D.J., N.A. and M.A.A.; funding acquisition, A.D.J. All authors have read and agreed to the published version of the manuscript.

**Funding:** This research was funded by the University College Dublin (UCD), School of Computer Science, Dublin, Ireland, grant number 13/RC/2077.

**Informed Consent Statement:** This article does not contain any studies with human participants or animals performed by any of the authors.

**Data Availability Statement:** Data in this research paper will be shared upon request made to the corresponding author.

**Conflicts of Interest:** All authors declare that they have no conflict of interest for the presented work.

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
