# Peer review of "A New Scheme for Ransomware Classification and Clustering Using Static Features"

_electronics, doi:10.3390/electronics11203307_

Round 1

Reviewer 1 Report

The paper claims to present a new scheme for ransomware classification. This "new scheme" is described in chapter 4 "Experimental Work", but this chapter contains only one (1!) page of text (to be exact: 52 lines) and 3 1/2 pages of more or less significant figures and tables. To give an example for "less significant":  table 6 contains the pseudocode for a program to scan a drive to find the biggest file there. This problem is an exercise in an elementary course for programming novices. It is hardly something that is appropriate for a paper in a scientific journal. It is not possible to find out, what the relevant innovative contribution of the paper is. In the whole paper, even in the abstract and the introduction, there is a permanent mix-up of concepts like detection, distinction between malware and benign code, classification, mitigation, decryption and the like. A clear definition of the relevant concepts used in the paper is missing. And if there is something like definition it reads like "Malware is malicious software or code that is designed to perform malicious actions." This is a circular definition which does not add any information to the word itself. It therefore does not match the most simple requirements of scientific work.

Moreover there are many details that make the paper hard to read (as one example: in line 60 the words "obfuscated and polymorphic hardware" introduces something I cannot even imagine. The paragraph below table 1 is completely unclear, if not understandable at all. How is ref. [10] (which is about using different ways of machine learning for API calls analysis) related to the rest of the paragraph, which does not mention machine learning at all? This paragraph is an example for the problem that characterizes the whole paper: a lot of concepts is mixed up without ever defining the differences between these concepts.

In the contributions at the end of chapter 1 a lot of questions remain:

- in point 1 "compare different ransomware families' infection behaviour": where do you get the ransomware and the ransomware families from? (a reference for the SALAM is necessary anyway).

- point 2 ends with an incomplete sentence.

- point 3 reads "Eliminate the limitation of previous research by finding the biggest file on the infected machine by developing an automatic script." This is ridiculous! Finding the biggest file on a machine is really trivial and cannot be a solution to anything.

- point 6 mentions a similarity matrix without giving any hint to what is compared.

In section 2.1 there is a "summary" of table 2- I cannot find any relation between this summary and the table. Moreover, table 2 is another example of the paper's general problem: It is completely unstructured with respect to the concepts mentioned and does not make any sense.

In line 184 you say you focus on the Salsa 20 encryption algorithm. If you do so (which is generally acceptable) you must explain why you opt for this restriction and what consequences this restriction has for the generality of your findings. At the end of the paragraph between figure 4 and figure 5 (lines 188-205) again there is a complete mixture of concepts: it talks about the Salsa 20 algorithm and then ends with social engineering. These two things really have nothing in common and the whole paragraph is just confusing. Figure 5 contains the pseudocode of an algorithm that again is just an exercise for programming beginners and has nothing to do with scientific research.

In section 2.3 again there are things put together that do not belong together and can therefore not be compared.

In section 2.4 you claim that "half of the ransomware victims do not recover their infected data after paying the ransom". How do you know, that it is 50%? Either you mention a reference or you have to describe, how you arrive at this number; anything else does not comply with minimal scientific standards. By the way: encrypted data is not equal to infected data.

Line 241 ff says "There are two types of ransomware decryption: one encrypts the victim's data and the other locks the victim's computer screen." I cannot imagine how locking a computer screen would decrypt anything.

After that paragraph there is a numbering problem (7. and 8. appear unexpectedly in the text).

By the end of chapter 2 I will stop with the notes on details. But it is not getting better at all. Still undefined concepts are used and things are compared that cannot be compared  (to compare "honeypots" and "Machine Learning" for example is not even comparing apples with oranges, but more like comparing the collection of mushrooms with different methods of cooking them!)

Chapter 4 does not state what you really did (again "developing a script that finds the largest file on a windows machine" is first of all not a scientific problem and, moreover, the code presented in figure 17 only finds the largest file on one drive of a windows machine!)

The conclusions claim things that are not described in the paper.

I am sorry, but this paper does not comply with minimal requirements for a scientific paper and should not be published. As it is a general problem, improvements on details would not change the overall impression.

Author Response

  1. Response to Comments and Suggestions from Reviewer 1

Reviewer#1, Concern #1:

1-The paper claims to present a new scheme for ransomware classification. This "new scheme" is described in chapter 4 "Experimental Work", but this chapter contains only one (1!) page of text (to be exact: 52 lines) and 3 1/2 pages of more or less significant figures and tables. To give an example for "less significant": table 6 contains the pseudocode for a program to scan a drive to find the biggest file there. This problem is an exercise in an elementary course for programming novices. It is hardly something that is appropriate for a paper in a scientific journal. It is not possible to find out, what the relevant innovative contribution of the paper is. In the whole paper, even in the abstract and the introduction, there is a permanent mix-up of concepts like detection, distinction between malware and benign code, classification, mitigation, decryption and the like. A clear definition of the relevant concepts used in the paper is missing. And if there is something like definition it reads like "Malware is malicious software or code that is designed to perform malicious actions." This is a circular definition which does not add any information to the word itself. It therefore does not match the most simple requirements of scientific work.

Author response:  We agree that Experimental work need more details, regarding pseudocode in figure 17 we agree won’t add value to the experimental work in the journal so we removed it after updating the whole Experimental work section, the contribution of the paper is to find a ransomware weakness in their encryption mechanism so we developed a new scheme to find similar ransomware samples using Jaccard Index depending on static features extraction and we successfully found a similar ransomware sample to our reference sample and the decryption tool decrypted the new sample encrypted files. We agree that Abstract and Introduction may miss clear definition of the relevant concepts, so we updated those two sections.  

Author action: We have updated the manuscript by removing figure 17 and updating Experimental work (4), Abstract and Introduction (1) sections.

2- Moreover there are many details that make the paper hard to read (as one example: in line 60 the words "obfuscated and polymorphic hardware" introduces something I cannot even imagine. The paragraph below table 1 is completely unclear, if not understandable at all. How is ref. [10] (which is about using different ways of machine learning for API calls analysis) related to the rest of the paragraph, which does not mention machine learning at all? This paragraph is an example for the problem that characterizes the whole paper: a lot of concepts is mixed up without ever defining the differences between these concepts.

Author response: We agree that Introduction section not clear, so we divided it to sub sections and reviewed and updated reference number 10.

Author action: We have updated the manuscript by updating Introduction (1) section, and we added two new subsections Ransomware Tracking Approaches (1.1) and Paper contributions (1.2)

Reviewer#1, Concern #2:

3- In the contributions at the end of chapter 1 a lot of questions remain: in point 1 "compare different ransomware families' infection behaviour": where do you get the ransomware and the ransomware families from? (a reference for the SALAM is necessary anyway).

Author response: The ransomware families collected through VirusTotal LiveHunting over 3 months, we mentioned this in Experimental work section Table 6. SALAM Ransomware reference is the reference number 11 in references section.

Author action: We answered the questions highlighted by the reviewer.

3- In the contributions at the end of chapter 1 a lot of questions remain: point 2 ends with an incomplete sentence.

Author response: We agree on reviewer comment, and we updated this point.

Author action: We have updated the manuscript by updating Paper contribution sub-section (1.2).

4- In the contributions at the end of chapter 1 a lot of questions remain: point 3 reads "Eliminate the limitation of previous research by finding the biggest file on the infected machine by developing an automatic script." This is ridiculous! Finding the biggest file on a machine is really trivial and cannot be a solution to anything.

Author response: We agree on reviewer comment, and we removed this point.

Author action: We have updated the manuscript by updating Paper contribution sub-section (1.2).

5- In the contributions at the end of chapter 1 a lot of questions remain:  point 6 mentions a similarity matrix without giving any hint to what is compared.

Author response: We agree on reviewer comment, and we removed this point.

Author action: We have updated the manuscript by updating Paper contribution sub-section (1.2).

Reviewer#1, Concern #3:

6- In section 2.1 there is a "summary" of table 2- I cannot find any relation between this summary and the table. Moreover, table 2 is another example of the paper's general problem: It is completely unstructured with respect to the concepts mentioned and does not make any sense.

Author response: Section 2.1 discuss Ransomware types and timeline history, so in table 2 we compared between different ransomware types and their infection routines, but we agree that section 2.1 need some improvement, so we changed the summary of section 2.1.

Author action: We have updated the manuscript by updating Ransomware Types and History section (2.1).

7- In line 184 you say you focus on the Salsa 20 encryption algorithm. If you do so (which is generally acceptable) you must explain why you opt for this restriction and what consequences this restriction has for the generality of your findings. At the end of the paragraph between figure 4 and figure 5 (lines 188-205) again there is a complete mixture of concepts: it talks about the Salsa 20 algorithm and then ends with social engineering. These two things really have nothing in common and the whole paragraph is just confusing. Figure 5 contains the pseudocode of an algorithm that again is just an exercise for programming beginners and has nothing to do with scientific research.

Author response: We agree with the reviewer comment, we updated section 2.2, and we removed figure 5.

Author action: We have updated the manuscript by updating Ransomware Encryption Mechanisms section (2.2).

8- In section 2.3 again there are things put together that do not belong together and can therefore not be compared.

Author response: We agree with the reviewer comment, we updated section 2.

Author action: We have updated the manuscript by removing section 2.3.

9- In section 2.4 you claim that "half of the ransomware victims do not recover their infected data after paying the ransom". How do you know, that it is 50%? Either you mention a r…

Author response: We agree with the reviewer comment, we updated section 2.

Author action: We have updated the manuscript by removing section 2.4.

Reviewer 2 Report

The authors surveyed the ransomware detection approaches and techniques to propose a scheme for ransomware classification and clustering using static features, which seems to provide a comprehensive survey for ransomware mitigation. However, there are some concerns which are as follows:

Figure 3 is exactly copied from the paper "https://www.mdpi.com/2071-1050/14/1/8". This is not advisable for any researcher to do the same (copy figure from other source). Authors must add its own contributions in the figure by changing the contents in the figure. Update this figure, otherwise Editor can reject this paper.

Also some contents are copied and not extended to till date (2022). The related work section does not seem to be comprehensive. The authors can add recent papers of 2021 and 2022 in ransomware detection approaches, ransomware detection ecosystem, and ransomware detection prevention recommendations. The authors can strengthen the comparative analysis table of different ransomware detection approaches by adding more number of recent papers.

In experimental work section, How authors validate the working of ransomware detection by studying dynamic features of different considered malware samples?

Quality of figure 4 is very poor. It looks like a screenshot is taken and updated in the paper. Diagrams must be originally designed.

The authors should recheck the numbering of subsection 7 and 8 which is entitled as "Screen-Locking Ransomware" and "Encryption Ransomware Decryptors". They do not seem to be correct.
Additionally, more information should be provided on these subsection.

Figure 11 is not visible clearly. The authors should improve visualization and quality of the figure.

The authors have mentioned ransomware detection tools and input information used in Figure 14. The authors did not discuss the detection tools by giving insights into the input information.

Author Response

Reviewer#2, Concern # 1:

1- Figure 3 is exactly copied from the paper "https://www.mdpi.com/2071-1050/14/1/8". This is not advisable for any researcher to do the same (copy figure from other source). Authors must add its own contributions in the figure by changing the contents in the figure. Update this figure, otherwise Editor can reject this paper.

Author response:  We agree with the reviewer that the Figure copied from the paper mentioned but we already added it the references, as recommended from the reviewer we updated the figure to our own contributes by changing the contents.

Author action: We updated the manuscript by updating the figure to our own contributes by changing the contents.

Reviewer#2, Concern # 2:

2-Some contents are copied and not extended to till date (2022). The related work section does not seem to be comprehensive. The authors can add recent papers of 2021 and 2022 in ransomware detection approaches, ransomware detection ecosystem, and ransomware detection prevention recommendations. The authors can strengthen the comparative analysis table of different ransomware detection approaches by adding more number of recent papers. 

Author response:  We updated Related work section and Comparison between ransomware detection approaches table, we have focused to demonstrate recent ransomware detection approaches, ransomware detection ecosystem, and ransomware detection prevention recommendations.

Author action: We updated the manuscript by updating Related work section (3) and updating table (5) by adding recent papers on ransomware detections approaches.

Reviewer#2, Concern # 3:

3- In experimental work section, how authors validate the working of ransomware detection by studying dynamic features of different considered malware samples?

Author response:  This paper experimental work focus on static features classification and clustering using similarity matrix of the Jaccard index for different features like (strings, import address table), regarding dynamic features we already working on dynamic features classification and clustering using APIs calls because we face a limitation on classify and cluster the packed samples with static features.

Author action: We updated the manuscript by adding sub-section on dynamic features discussion on the experimental work section.

Reviewer#2, Concern # 4:

4- Quality of figure 4 is very poor. It looks like a screenshot is taken and updated in the paper. Diagrams must be originally designed.

Author response:  We agree with the reviewer that quality of figure 4 is very poor, we updated the figure by our design.

Author action: We updated the manuscript by updating figure 4 with good quality figure by our design.

Reviewer#2, Concern # 5:

5- The authors should recheck the numbering of subsection 7 and 8 which is entitled as "Screen-Locking Ransomware" and "Encryption Ransomware Decryptors". They do not seem to be correct. Additionally, more information should be provided on these subsection.

Author response:  We agree with the reviewer the subsections 7 and 8 not correct.

Author action: We updated the manuscript by removing subsection 7 and 8 from the background.

Reviewer#3, Concern # 6:

6- Figure 11 is not visible clearly. The authors should improve visualization and quality of the figure.

Author response:  We agree with the reviewer that figure 11 not visible clearly, we updated the figure by our design.

Author action: We updated the manuscript by updating figure 11 with good quality figure by our design.

Reviewer#3, Concern # 7:

7- The authors have mentioned ransomware detection tools and input information used in Figure 14. The authors did not discuss the detection tools by giving insights into the input information.

Author response:  We agree with the reviewer that we didn’t give enough insights on detection tools and input information that mentioned by Related work paper authors.

Author action: We updated the manuscript by updating Related work section (3) and Ransomware Detection Ecosystem subsection (3.2)

Round 2

Reviewer 1 Report

I see some improvements in the manuscript. Still the main problem remains: you generally mix-up concepts that do not belong together and you do not clearly define the concepts used. For example it is still unclear whether for you there is a difference between malware and ransomware or do you use the 2 terms synonymously?

As an example for the mentioned mix-up, see section 2.1: 

Example 1: "Ransomware can be divided into different types: Crypto worm [12], Ransomware-as-a-Service (RaaS) [13], and Automated Active Adversary"

These 3 terms describe quite different properties of malware are not useful for classification of ransomware (RaaS just says where you get the ransomware from, while worm is a special way of malware distribution.

Example 2:

"The sources of infection for most 140 ransomware ... can be summarized as follows.

• Phishing emails. 

• APT attacks 

• System vulnerabilities (RDP)

• Drive-by downloads

• Exploit kits"

To my understanding the only infection sources are phishing emails and drive-by-downloads; system vulnerabilities are the targets of an infection; exploit kits are tools to exploit vulnerabilities; and APT (Advanced Persistent Threat) is a very general and vague classification for sophisticated malware.

A clear definition what you mean by the term feature is still missing.

Section 2.2: Where did you get the information of Table 4 from? Did you find out yourself? And if yes, how did you do it? By the way: Table 4 does not summarize the different ransomware techniques; it summarizes only the different encryption techniques used.

Section 2.3  "Static Features Malware Tracking" starts with the following sentence: "This system aims to help malware analysts ...". Which system???

Again in section 2.3 there is a mix-up of the terms "features", "ng-rams" and "events".

The paragraph inserted at the end of section 3.2 does not clarify anything. I just do not understand what you mean.

The additional page inserted in section 4 does not really help to understand what you really did and how you did it.

Overall the revised paper has improved a bit, but to my opinion it still does not fulfil the requirements of a sound scientific paper.

Author Response

  1. Response to Comments and Suggestions from Reviewer 1

Reviewer#1, Concern #1:

1- generally mix-up concepts that do not belong together and you do not clearly define the concepts used. For example, it is still unclear whether for you there is a difference between malware and ransomware, or do you use the 2 terms synonymously?

Author response:  We agree on reviewer comment, we focus on ransomware analysis, and we tried to use ransomware term as possible as we can, we added a new sub section to explain that Ransomware a sub category of malwares.

Author action: We have updated the manuscript by adding sub section Malware Via Ransomware (1.1) section.

Reviewer#1, Concern #2:

2- Example 1: "Ransomware can be divided into different types: Crypto worm [12], Ransomware-as-a-Service (RaaS) [13], and Automated Active Adversary"

These 3 terms describe quite different properties of malware are not useful for classification of ransomware (RaaS just says where you get the ransomware from, while worm is a special way of malware distribution.

Author response: We explained more about ransomware crypto-worm capabilities as it legacy type of the ransomwares used by different ransomware families like WannaCry crypto-worm, we added more information about the Idea of Ransomware as a service (RaaS) which involved a lot of parties in ransomware delivery, the ransomware developer who sell it, the attacker who initiate the attack using this ransomware and share profits, regarding Automated Active Adversary ransomware which used by APT (Advanced persistent threat) groups is a targeted attack in order to destroy the environment.

Author action: We have updated the manuscript by updating Ransomware Types and History (2.1) section.

Reviewer#1, Concern #3:

3- Example 2:

"The sources of infection for most ransomware ... can be summarized as follows.

  • Phishing emails.
  • APT attacks
  • System vulnerabilities (RDP)
  • Drive-by downloads
  • Exploit kits

To my understanding the only infection sources are phishing emails and drive-by-downloads; system vulnerabilities are the targets of an infection; exploit kits are tools to exploit vulnerabilities; and APT (Advanced Persistent Threat) is a very general and vague classification for sophisticated malware.

A clear definition what you mean by the term feature is still missing.

Author response: We added new sub section to explain different ransomware infection sources, as attackers used to use random phishing emails and Drive-by downloads but modern ransomware infection source is human-operated ransomware which is the result of a targeted attack by cybercriminals who gain access to an organization's on-premises or cloud IT infrastructure, gain administrative access, and then use that access to push ransomware to critical data via domain controller group policy or network shares folders. Human-operated ransomware attacks frequently include stealing credentials and then using those credentials to move laterally within an organization and get administrative access to more accounts. It's possible for fraudsters to exploit security configuration gaps during maintenance windows. The end goal is to release a malware payload into the system. We added new table to differentiate between ransomware infection source types.

Author action: We have updated the manuscript by new sub-section Ransomware Infection source routine (2.2) section and table 3.

Reviewer#1, Concern #4:

4- Section 2.2: Where did you get the information of Table 4 from? Did you find out yourself? And if yes, how did you do it? By the way: Table 4 does not summarize the different ransomware techniques; it summarizes only the different encryption techniques used.

Author response: We agree on reviewer comment, and we changed the table name to Ransomware Encryption Algorithms. That information we got from our previous research in references [11] and [53], for other encryption algorithm types we mentioned the other references [18] [26] [54].

Author action: We have updated the manuscript by updating the table and adding new references.

Reviewer#1, Concern #5:

5- Section 2.3 "Static Features Malware Tracking" starts with the following sentence: "This system aims to help malware analysts ...". Which system???

Author response: We agree on reviewer comment, and we explained this point, this system is our proposed ransomware classification, clustering, and detection system in Experimental Work section

Author action: We have updated the manuscript by updating Static Features Ransomware Tracking sub-section (2.3).

Reviewer#1, Concern #6:

6- Again in section 2.3 there is a mix-up of the terms "features", "ng-rams" and "events".

Author response: We agree on reviewer comment, we updated Static Features Ransomware Tracking sub section by explain n-grams in more clear way mentioned in previous electronics journal reference [56], and we added new figure to explain n-grams extracted from ransomware samples.

Author action: We have updated the manuscript by updating Static Features Ransomware Tracking sub-section (2.3) and added new figure 6.

Reviewer#1, Concern #7:

7- The paragraph inserted at the end of section 3.2 does not clarify anything. I just do not understand what you mean.

Author response: We agree with the reviewer comment, we updated the paragraph to be clearer, as the authors of this related work provide a categorization of the data and metrics gleaned from ransomware activity. They have something to do with the many forms of ransomware. In the first level of categorization, they grouped over 16 parameters related to prior actions into 3 groupings. They distinguished static or dynamic data gathered locally from the infected machine from data gathered through the network. In the parts that follow, they examine the following three groups: The information is locally static, meaning it is gathered prior to running the malware program by extracting it. Second, data is gathered locally in real time as the malware operates on the infected machine. Thirdly, information is gleaned through the malware's own network activity.

Author action: We have updated the manuscript by updating Ransomware Detection Ecosystem section (3.2).

Reviewer#1, Concern #8:

8- The additional page inserted in section 4 does not really help to understand what you really did and how you did it.

Author response: We added this page according to second reviewer recommendations.

Author action: We have updated the manuscript by updating Experimental Work section (4).

Reviewer 2 Report

The authors have updated the manuscript, so no further comments. It is now in acceptable form.

Author Response

  1. Response to Comments and Suggestions from Reviewer 2

Recommendation: Accept (The authors have updated the manuscript, so no further comments. It is now in acceptable form)

Response:

The authors greatly appreciate that the reviewer has found the current version acceptable for the Electronics 2022 Journal.

Round 3

Reviewer 1 Report

The changes have improved the paper considerably. Removal of typos and editing of English language style is still necessary.

Author Response

The authors greatly appreciate that the reviewer has found the current version acceptable for the Electronics 2022 Journal. We further improved the English style, and we hope the current version satisfies the reviewer's needs.